# Reinforcement Fine-Tuning Naturally Mitigates Forgetting in Continual Post-Training

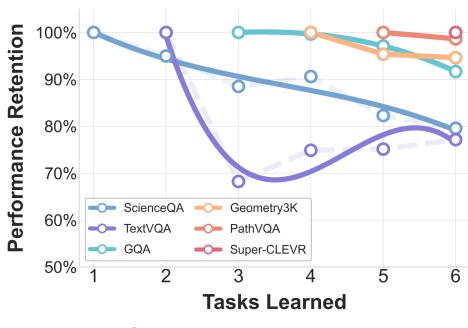
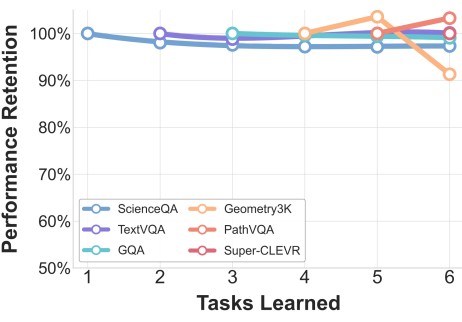

**(a) Supervised Fine-tuning**  **(b) Reinforcement Fine-tuning**

Figure 1: Comparison of performance retention between SFT and RFT in continual post-training. We plot the performance on each task, normalized relative to its initial post-training peak, as the model learns through a sequence of multimodal tasks. **(a)** SFT exhibits classic catastrophic forgetting, where performance on previously learned tasks degrades dramatically as new tasks are introduced. **(b)** By contrast, RFT demonstrates remarkable stability, maintaining high performance on prior tasks throughout the entire sequence. This suggests an inherent forgetting-mitigation property within the RFT paradigm. Further details on the experimental setup can be found in Section 4.

## Abstract

Continual post-training (CPT) is a popular and effective technique for adapting foundation models like multimodal large language models to specific and ever-evolving downstream tasks. While existing research has primarily concentrated on methods like data replay, model expansion, or parameter regularization, the fundamental role of the learning paradigm within CPT remains largely unexplored. This paper presents a comparative analysis of two core post-training paradigms: supervised fine-tuning (SFT) and reinforcement fine-tuning (RFT), investigating their respective impacts on knowledge retention during CPT. Our experiments are conducted on a benchmark comprising seven diverse multimodal tasks, utilizing Qwen2.5-VL-7B-Instruct as the base model for continual post-training. The investigation yields two significant findings: (1) When continuously learning on downstream tasks, SFT leads to catastrophic forgetting of previously learned tasks. In contrast, RFT inherently preserves prior knowledge and achieve performance comparable to multi-task training. (2) RFT successfully protects and even enhances the model's general knowledge on standard benchmarks (e.g., MMMU and MMLU-Pro). Conversely, SFT degrades general model capabilities severely. Further analysis reveals that this stability is not primarily due to explicit mechanisms like KL penalty or chain-of-thought reasoning. Instead, we identify an implicit regularization mechanism inherent to RFT as a key contributing factor. Our theoretical analysis suggests that RFT's gradient updates are naturally scaled by the reward variance, acting as a data-dependent regularizer that inherently protects previously acquired knowledge. Finally, we propose a rollout-based instance filtering algorithm to enhance the stability and efficiency of RFT. Our comprehensive study demonstrates the superiority of RFT as a robust paradigm for continual post-training. [1]

---

[1] Our code is provided in the supplementary material. An anonymous link for review is: https://anonymous.4open.science/r/RFTvsSFT-A999

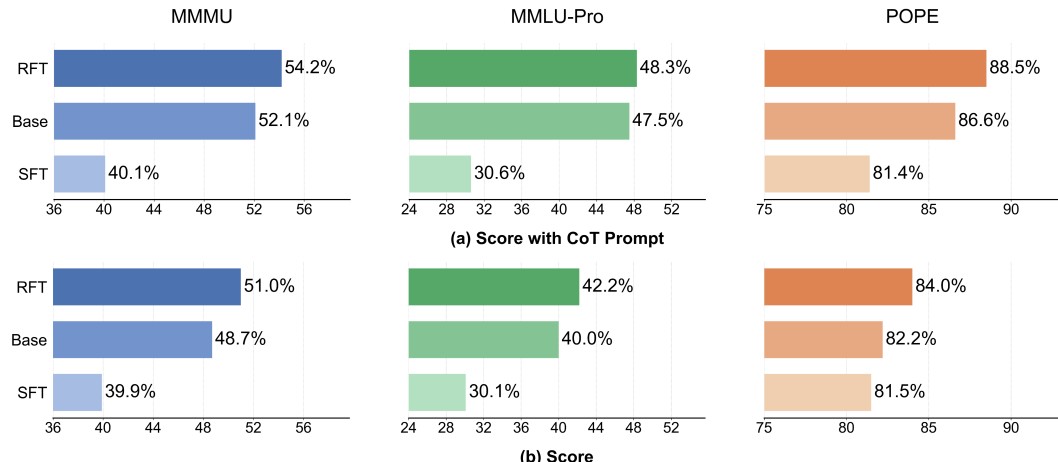

Figure 2: General capability preservation after continual post-training. We evaluate models at the end of learning all downstream tasks on general benchmarks using both CoT and direct prompting. Compared to the base model, SFT (shown in light colors) causes degradation while RFT (shown in darker colors) preserves and even enhances general capabilities.

# 1 INTRODUCTION

Recent advancements in multimodal large language models (MLLMs) have demonstrated remarkable capabilities in complex world understanding (Achiam et al., 2023; Liu et al., 2024; Wang et al., 2024a). To align with the demands of real-world deployment, MLLMs must adapt to a stream of data and evolving user requirements, incorporating new skills and domain knowledge over time (Zhu et al., 2024). This calls for an efficient and scalable continual post-training (CPT) paradigm. A key challenge in CPT is the well-known phenomenon of catastrophic forgetting (McCloskey & Cohen, 1989), where adapting to a new task leads to a severe degradation of performance on previously learned tasks. To reduce forgetting, recent studies (Guo et al., 2025c) focus on data replay (Maharana et al., 2025; Lee et al., 2025; Wang et al., 2025), model expansion (Zhao et al., 2025; Guo et al., 2025b; Zeng et al., 2024), and explicit regularization (Liu et al., 2025a). Nevertheless, existing methods typically leverage the supervised fine-tuning (SFT) paradigm by default, and the role of the fundamental fine-tuning paradigm in CPT has been overlooked.

Recently, reinforcement fine-tuning (RFT), which optimizes models based on feedback from generated outputs, has significantly advanced foundation model post-training (Chu et al., 2025; Shao et al., 2024; Guo et al., 2025a). To the best of our knowledge, this work presents the first direct comparative investigation into whether SFT or RFT is the more suitable paradigm for CPT, focusing on knowledge preservation for both specific downstream tasks and general capabilities. Experimentally, we continually fine-tune the Qwen2.5-VL-7B-Instruct model (Bai et al., 2025c) on a benchmark comprising diverse multimodal tasks covering various domains. To fully reflect the knowledge preservation ability, we evaluate forgetting on both learned specific tasks and general benchmarks such as MMMU (Yue et al., 2024), MMLU-Pro (Wang et al., 2024b), and POPE (Li et al., 2023a).

The empirical investigation yields two notable findings: (**1**) As shown in Figure 1, when continuously learning on downstream tasks, SFT leads to catastrophic forgetting of previously learned tasks, which is consistent with existing studies (Guo et al., 2025c). In contrast, RFT can inherently protect prior knowledge, maintaining strong performance on old tasks after being adapted to new tasks. Surprisingly, without any data replay, continual post-training with RFT can achieve comparable performance with that of multi-task training, which is not achievable even when equipping SFT with continual learning strategies. (**2**) As demonstrated in Figure 2, continual training on downstream tasks with SFT severely degrades general model capabilities, which is known as base model degradation (Liu et al., 2025a). For example, the performance drops from 52.1% to 40.1% on MMMU. Fortunately, RFT protects the general performance and enhances the model's general knowledge (52.1% → 54.2%). These observations highlight the knowledge preservation capability of RFT.

To understand how RFT mitigates forgetting during CPT, we conduct additional experiments with the popular and representative group relative policy optimization (GRPO) framework (Shao et al., 2024). We analyze the impact of KL divergence penalty and chain-of-thought (CoT) reasoning (Wei et al., 2022) on forgetting mitigation. Particularly, the KL divergence penalty prevents the policy from changing too drastically, similar to the well-known knowledge distillation in continual learning (Li & Hoiem, 2017). However, our analysis indicates that these explicit mechanisms are not the primary drivers of forgetting mitigation. We instead attribute this phenomenon to an implicit regularization effect within RFT. We offer a theoretical perspective suggesting that RFT's updates are inherently more conservative in parameter subspaces sensitive to prior tasks. This conservatism is naturally scaled by the variance of the reward signal, creating a data-dependent regularization that dampens updates on uncertain samples, thus protecting established knowledge. Last but not least, we observe that the learning process of RFT can be highly inefficient. Thus, we introduce a rollout-based instance filtering algorithm that enhances the stability of GRPO while still being an excellent knowledge protector.

Our main contributions are threefold:

1. We present the first comprehensive analysis of the forgetting mitigation effects of SFT and RFT during continual post-training of MLLMs, demonstrating that RFT naturally preserves not only the performance of learned downstream tasks but also general model capabilities.

2. Based on in-depth analyses, we reveal that the implicit regularization introduced by RFT significantly contributes to the forgetting mitigation, being more important than KL regularization and CoT reasoning.

3. We propose a rollout-based instance filtering algorithm that enhances the stability and efficiency of RFT while still maintaining previous learned knowledge.

## 2 RELATED WORKS

**Continual Post-Training in MLLMs.** Continual learning aims to enable models to learn from a stream of tasks without catastrophically forgetting previously acquired knowledge (Van de Ven et al., 2022). For MLLMs, this capability is particularly important for adapting these powerful models to a diverse range of downstream multimodal tasks. Existing CPT research in MLLMs (Guo et al., 2025c) has focused on adapting traditional forgetting mitigation strategies such as regularization, data replay, and model expansion, within an SFT paradigm. Regarding benchmark, Chen et al. (2024) introduced a continual instruction tuning benchmark including several specific multimodal datasets. Zhao et al. (2025) introduces two settings named domain continual learning and ability continual learning, providing a realistic evaluation for continual post-training of MLLMs. In addition to these methods, recent efforts to mitigate catastrophic forgetting in MLLMs primarily focus on parameter-efficient learning and dynamic data selection. For instance, HiDe-LLaVA (Guo et al., 2025b) employs a hierarchical decoupling framework for task-specific LoRA expansion and general knowledge fusion. MRLoRA (Zhao et al., 2025) leverages architectural decoupling and a multimodal routing mechanism to selectively activate specialized parameters. In terms of data management, Adapt-∞ (Maharana et al., 2025) dynamically selects high-impact samples based on gradient representations and prunes redundant data. These diverse strategies collectively aim to enhance the ability of MLLMs to continually learn new tasks while preserving previously acquired knowledge. Recently, Liu et al. (2025a) developed LLaVA-c, which is a simple yet effective CPT framework for MLLMs, addressing task balancing and catastrophic forgetting through spectral-aware consolidation and unsupervised inquiry regularization.

**Post-Training of Foundation Models.** Post-training is a critical stage for refining the capabilities of pre-trained foundation models (Shao et al., 2024; Chu et al., 2025; Achiam et al., 2023). SFT on task-specific or instruction-formatted datasets is a common approach to adapt models to downstream applications (Chung et al., 2024; Zhou et al., 2023). For example, Chung et al. (2024) demonstrated that by scaling the number of tasks and model size, and incorporating CoT data, SFT significantly enhances the performance and generalization of various large language models across diverse benchmarks. Recently, RFT has gained prominence for aligning models with human preferences or improving performance on specific objectives (Liu et al., 2025c; Zhai et al., 2024; Shao et al., 2024; Luong et al., 2024; Li et al., 2025; 2023c; Ahmadian et al., 2024). Particularly, GRPO

(Shao et al., 2024) largely enhances mathematical reasoning and optimizes memory usage, being a popular method for post-training of large language models. Liu et al. (2025b) revealed inherent biases in the GRPO algorithm, then introduces an unbiased optimization method that improves token efficiency while maintaining reasoning performance. Visual-RFT (Liu et al., 2025c) boosts MLLMs by using reinforcement learning with rule-based visual rewards, making them more data-efficient and better at various visual tasks than traditional SFT. Recently, Chu et al. (2025) demonstrated that reinforcement learning significantly enhances the generalization capabilities of foundation models, while SFT primarily leads to memorization. In this work, we study the comparative effect of SFT and RFT on knowledge retention in MLLMs continual post-training. Recent work by Zhang et al. (2025) investigates SFT and RFT from a data perspective, showing that incorporating reasoning trajectories in SFT can reduce forgetting. Their findings complement our work by highlighting how data format affects SFT's stability, while we demonstrate that RFT provides inherent forgetting mitigation without reasoning format. Together, these studies provide comprehensive guidance for post-training paradigm selection.

## 3 PRELIMINARIES

Post-training is a critical phase following large-scale pre-training that adapts foundation models to specific downstream tasks or align them with human preferences (Ouyang et al., 2022; Kumar et al., 2025). We model the MLLM with parameters $\theta$ as a policy $\pi_\theta$. This policy defines a conditional probability distribution $\pi_\theta(a|x)$ over possible text responses $a$ given a multimodal input prompt $x$, which consists of text and images. We also assume a scalar reward function $r(x, a) \in \mathbb{R}$ that evaluates the quality of a response. Post-training aims to update the parameters $\theta$ of a pre-trained base model $\pi_{\theta_{\text{base}}}$ to improve its performance on a downstream task using a training dataset $\mathcal{D}$, which can be achieved by SFT (Ouyang et al., 2022) or RFT (Lee et al., 2023).

**SFT.** Given training dataset $\mathcal{D} = \{(x_i, a_i^*)\}_{i=1}^N$ consisting of prompts $x_i$ and their corresponding ground-truth responses $a_i^*$, SFT maximizes the likelihood of generating the ground-truth responses. This is typically achieved by minimizing the negative log-likelihood loss:

$$\mathcal{L}_{\text{SFT}}(\theta) = -\mathbb{E}_{(x,a^*)\sim\mathcal{D}}[\log \pi_\theta(a^*|x)] = -\mathbb{E}_{(x,a^*)\sim\mathcal{D}}\left[\sum_{t=1}^{|a^*|} \log \pi_\theta(a_t^*|x, a_{<t}^*)\right]. \quad (1)$$

**RFT.** In RFT, the model $\pi_\theta$ is treated as a policy, and generates one or more candidate responses for a given prompt $x$. The optimization objective is to maximize the expected reward:

$$\mathcal{J}_{\text{RFT}}(\theta) = \mathbb{E}_{x\sim\mathcal{D}}\mathbb{E}_{a\sim\pi_\theta(\cdot|x)}[r(x, a)]. \quad (2)$$

The gradient of this objective is typically estimated using policy gradient methods. The most basic form is the REINFORCE (Williams, 1992) estimator, which, unfortunately, has high gradient variance. Recent RFT algorithms (Shao et al., 2024; Li et al., 2023c; Ahmadian et al., 2024) address this issue by designing more stable advantage estimators and baselines. We introduce some of the representative methods used in our study below.

For a prompt $x$, **GRPO** (Shao et al., 2024) generates a group of $n$ responses $\{a_1, \ldots, a_n\}$ and computes their rewards $\{r_1, \ldots, r_n\}$. The advantage for a response $a_i$ is its normalized reward relative to the group mean: $A(a_i) = (r_i - \bar{r})/\sigma_r$, where $\bar{r}$ and $\sigma_r$ are the mean and standard deviation of the rewards. The objective is to maximize the expected advantage-weighted log-probability, often with a KL-divergence penalty against a reference policy $\pi_{\text{ref}}$ to stabilize training:

$$\mathcal{J}_{\text{GRPO}}(\theta) = \mathbb{E}_{x,\{a_i\}}\left[\sum_{i=1}^n A(a_i) \log \pi_\theta(a_i|x)\right] - \beta D_{\text{KL}}(\pi_\theta(\cdot|x)||\pi_{\text{ref}}(\cdot|x)), \quad (3)$$

where $\beta > 0$. **ReMax** (Li et al., 2023c) use the reward of a greedy decoding response $\hat{a}$ as a baseline. For a single sampled response $a$, the objective is to maximize:

$$\mathcal{J}_{\text{ReMax}}(\theta) = \mathbb{E}_{x,a\sim\pi_\theta}\left[(r(x, a) - r(x, \hat{a})) \log \pi_\theta(a|x)\right]. \quad (4)$$

This adaptive baseline helps to normalize rewards and reduce gradient variance. To further reduce variance, **RLOO** (Ahmadian et al., 2024) generates $n$ samples $\{a_1, \ldots, a_n\}$ and uses the average

reward of the other $n-1$ samples as a baseline for sample $a_i$:

$$\mathcal{J}_{\text{RLOO}}(\theta) = \mathbb{E}_{x,\{a_i\}} \left[ \frac{1}{n} \sum_{i=1}^{n} \left( r(x,a_i) - \frac{1}{n-1} \sum_{j \neq i} r(x,a_j) \right) \log \pi_\theta(a_i|x) \right].$$ (5)

**Continual Post-Training Formulation.** In CPT, the model learns from a sequence of $T$ tasks with datasets $\{\mathcal{D}_1, \ldots, \mathcal{D}_T\}$. The core challenge is catastrophic forgetting, i.e., a significant drop in performance on previously learned tasks. Following the general continual learning framework, CPT can be formulated as a constrained optimization problem. When learning task $t$, the objective is:

$$\theta^t = \arg\min_\theta \mathcal{L}(\theta; \mathcal{D}_t) \quad \text{s.t.} \ \mathcal{L}(\theta; \mathcal{D}_i) \leq \mathcal{L}(\theta^i; \mathcal{D}_i), \quad \forall i \in [1, t-1]$$ (6)

where $\mathcal{L}(\theta; \mathcal{D}_i)$ is the training objective (e.g., negative log-likelihood for SFT or negative expected reward for RFT) on task $i$, and $\theta^i$ are parameters after learning task $i$.

## 4 REINFORCEMENT FINE-TUNING MITIGATES FORGETTING IN CPT

This section presents our comparative results comparing RFT and SFT in a continual post-training scenario. We detail our experimental setup and then present the main findings that highlight the superiority of RFT for knowledge preservation.

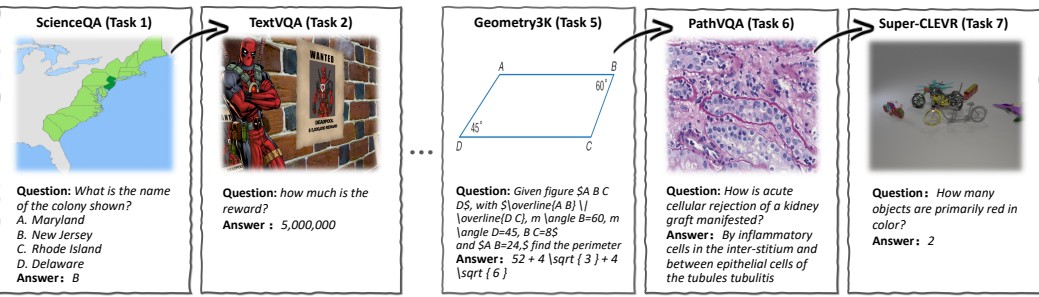

Figure 3: Illustrative examples of continual post-training benchmark.

### 4.1 EXPERIMENTAL SETUP

**Continual Post-Training Model & Datasets.** We adopt the open-source Qwen2.5-VL-7B-Instruct (Bai et al., 2025b) as our base model, primarily due to its demonstrated superiority in vision-language comprehension and its favorable resource footprint, which is crucial for practical deployment. We continually fine-tune the model on diverse vision-language datasets (ScienceQA (Saikh et al., 2022), TextVQA (Singh et al., 2019), VizWiz (Gurari et al., 2018), GQA (Hudson & Manning, 2019), Geometry3K (Lu et al., 2021), PathVQA (He et al., 2020), Super-CLEVR (Li et al., 2023b)), covering a wide range of common downstream applications. After the end of CPT, evaluation is performed on the test sets of all previously encountered tasks. Additionally, to fully assess the knowledge preservation ability, we evaluate the model on diverse, general benchmarks at the end of learning all downstream tasks. Specifically, we evaluate the model on three specialized benchmarks: MMMU (Yue et al., 2024), MMLU-Pro (Wang et al., 2024b), and POPE (Li et al., 2023a). Particularly, we include POPE to systematically assess whether CPT induces object hallucination in MLLMs. A detailed description of those datasets is provided in the Appendix A.

**Learning Algorithms & Reward.** Our experiments encompass a range of fine-tuning algorithms, including standard SFT (Zheng et al., 2024) and several representative RFT algorithms, i.e., GPRO (Shao et al., 2024), ReMax (Li et al., 2023c), and RLOO (Ahmadian et al., 2024). For both SFT and RFT, model outputs are normalized by disregarding extraneous whitespace (e.g., spaces, indentations, newlines) and ignoring case sensitivity to ensure precise assessment. For GRPO, the overall reward $r_{\text{overall}}$ is designed with a weighted sum of accuracy reward and format reward:

$$r_{\text{overall}} = 0.9 r_{\text{acc}} + 0.1 r_{\text{format}}.$$ (7)

Table 1: Final performance comparison on all tasks after the entire continual learning sequence. The **best** and second-best results are highlighted. "-" indicates that the metric is not applicable.

| Method | SciQA | TextVQA | VizWiz | GQA | Geo. | PathVQA | sCLEVR | AvgAcc | FM |
|--------|-------|---------|--------|-----|------|---------|--------|--------|-----|
| Base | 90.5 | 62.8 | 45.5 | 47.2 | 37.7 | 21.8 | 41.1 | 49.5 | - |
| MTL (SFT) | 95.2 | 69.9 | 64.5 | 63.4 | 18.1 | 61.6 | 57.5 | 62.9 | - |
| SFT | 76.1 | 55.8 | 46.8 | 58.5 | 20.2 | **62.2** | **58.2** | 54.0 | -10.4 |
| ReMax | 87.6 | 71.4 | 51.6 | 62.4 | 16.8 | 33.3 | 54.1 | 53.9 | -3.8 |
| RLOO | **94.0** | 73.7 | 48.9 | 62.7 | **42.1** | 40.5 | 55.3 | 59.6 | **-2.1** |
| GRPO | 93.0 | **74.8** | **51.8** | **65.9** | 38.4 | 41.3 | 54.2 | **60.0** | -2.3 |

Specifically, the accuracy reward $r_{\text{acc}}$ assesses the semantic correctness of the generated content, which yields a reward of 1 if the generated answer $a$ matches the ground truth answer $a^*$, and 0 otherwise. The format reward assesses adherence to the expected output structure. It utilized regular expressions to verify the correct presence and formatting of the CoT reasoning block, delineated by `<think>` and `</think>` tags, and the final answer encapsulated within a `\boxed{}` environment. A perfect format match resulted in a score of 1, otherwise 0.

**Prompt Template.** Our base model, Qwen-VL-7B-Instruct, utilizes two kinds of input prompt templates, as illustrated in the Appendix. *NoCoT* (non-chain-of-thought) prompt template adheres to a basic question-answering format, where the question text is presented directly, and the model is expected to provide the final answer without intermediate steps. Differently, in *CoT* prompt template, the query's question text is directly incorporated into the prompt, followed by an instruction for the model to first engage in a reasoning process. This CoT reasoning is then generated within a dedicated `<think>` and `</think>` block. The final answer is explicitly distinguished and encapsulated within a `\boxed{}` environment.

**Evaluation Metrics.** To quantify the model's performance during CPT, we adopt two standard metrics. Let $P_{t,j}$ denote the test accuracy on task $j$ after learning task $t$. We measure the final overall performance using ***average accuracy*** (***AvgAcc***), which is the average accuracy across all tasks after training on the final task $T$. To measure knowledge retention, we use the ***forgetting measure*** (***FM***), which calculates the average difference between the final accuracy of a task and the best accuracy achieved for that task throughout the training sequence. Let $P_i^* = \max_{k \in \{i,...,T\}} P_{k,i}$ be the best performance for task $i$. The above two metrics are defined as:

$$AvgAcc = \frac{1}{T}\sum_{i=1}^{T} P_{T,i}, \qquad FM = \frac{1}{T}\sum_{i=1}^{T}(P_{T,i} - P_i^*). \tag{8}$$

A higher *AvgAcc* indicates better overall performance, while an *FM* closer to zero signifies less forgetting and better knowledge preservation.

**Implementation Details.** All experiments employ full-parameter fine-tuning for both SFT and RFT to ensure comprehensive capability assessment. Experiments of SFT are conducted using the *llamafactory* (Zheng et al., 2024) framework, with a learning rate of $1e-5$ and a batch size of 24. RFT methods (GRPO, ReMax, and RLOO) are implemented using the *easyR1* (Zheng et al., 2025) framework, building upon *Verl* (Sheng et al., 2024). A consistent configuration is applied across RFT methods to ensure an equitable comparison: a learning rate of $1e-6$, a rollout batch size of 512, a sampling temperature of 1.0, with KL-divergence coefficient $\beta = 0.01$. Specifically, GRPO is implemented adhering to its foundational methodology, with a group size set to 8. ReMax followed its core algorithm, and RLOO adopted the official Hugging Face algorithm. To ensure the generality of our findings, we conduct additional experiments across different model architectures, scales, and task domains, with detailed results provided in Appendix D.

## 4.2 FINDING 1: RFT INHERENTLY RESISTS CATASTROPHIC FORGETTING

Our primary investigation focuses on the knowledge retention capabilities of SFT and RFT within a continual learning sequence. The results, summarized in Table 1, reveal a contrast between the two paradigms.

**SFT suffers from catastrophic forgetting.** We observe that sequential SFT leads to a severe degradation of performance on previously learned tasks with a forgetting measure (FM) of **-10.4%**. For instance, performance on ScienceQA drops dramatically (95.2% → 76.1%) after completing the entire task sequence. The final average accuracy (AvgAcc) of **54.0%** is substantially lower than the multi-task learning of SFT, which is the upper bound of **62.9%**, confirming that SFT is highly susceptible to forgetting.

Table 2: General capabilities evaluation on MMMU, MMLU-Pro, and POPE benchmarks after training on downstream tasks. The **best** and second-best results are highlighted.

| Benchmark | Eval CoT | Base | SFT | | RFT | | |
| --- | --- | --- | --- | --- | --- | --- | --- |
| | | | SFT | MTL (SFT) | GRPO | RLOO | ReMax |
| MMMU | ✔ | 52.1 | 40.1 (↓**12.0**) | 47.8 (↓**4.3**) | **54.2** (↑**2.1**) | 53.7 (↑**1.6**) | 48.7 (↓**3.4**) |
| | ✗ | 48.7 | 39.9 (↓**8.8**) | 48.1 (↓**0.6**) | 51.0 (↑**2.3**) | 46.8 (↓**1.9**) | **51.6** (↑**2.9**) |
| MMLU-Pro | ✔ | 47.5 | 30.6 (↓**16.9**) | 33.2 (↓**14.3**) | **48.3** (↑**0.8**) | 45.1 (↓**2.4**) | 35.4 (↓**12.1**) |
| | ✗ | 40.0 | 30.1 (↓**9.9**) | 32.9 (↓**7.1**) | **42.2** (↑**2.2**) | 39.7 (↓**0.3**) | 41.0 (↑**1.0**) |
| POPE | ✔ | 86.6 | 81.4 (↓**5.2**) | 84.9 (↓**1.7**) | **88.5** (↑**1.9**) | 88.2 (↑**1.6**) | 85.2 (↓**1.4**) |
| | ✗ | 82.2 | 81.5 (↓**0.7**) | 84.5 (↑**2.3**) | 84.0 (↑**1.8**) | 82.0 (↓**0.2**) | **87.2** (↑**5.0**) |

Table 3: Downstream task performance for ablation models. We investigate the role of the KL term and CoT through variants of GRPO. † indicates that the training process is unstable and requires multiple restarts from a previous checkpoint to achieve convergence.

| Method | SciQA | TextVQA | VizWiz | GQA | Geo. | PathVQA | sCLEVR | AvgAcc |
| --- | --- | --- | --- | --- | --- | --- | --- | --- |
| SFT | 76.1 | 55.8 | 46.8 | 58.5 | 20.2 | **62.2** | **58.2** | 54.0 |
| GRPO | 93.0 | 74.8 | 51.8 | 65.9 | **38.4** | 41.3 | 54.2 | **60.0** |
| GRPO w/o KL | 93.0 | **75.0** | 51.6 | 65.9 | $35.6^{\dagger}$ | $40.9^{\dagger}$ | $54.7^{\dagger}$ | 59.5 |
| GRPO w/o CoT | **94.7** | 74.7 | **63.8** | 65.9 | 23.8 | 38.2 | 54.4 | 59.4 |

**RFT preserves task knowledge and achieves MTL performance.** In contrast, all RFT methods demonstrate remarkable resilience against forgetting. As shown in Table 1, RFT methods exhibit very low forgetting measures, with GRPO achieving an FM of **-2.3%**. For example, GRPO maintains ScienceQA performance at **93.0%** after learning all tasks, compared to its peak performance of **95.6%**, which is a minimal drop compared to SFT. Among RFT methods, GRPO performs best, achieving a final AvgAcc of **60.0%**, which is close to the upper bound of **62.9%**. The model achieves this high performance *without any explicit continual learning strategies*, suggesting that the RFT paradigm is inherently robust for CPT.

### 4.3 FINDING 2: RFT PROTECTS AND ENHANCES GENERAL CAPABILITIES

Beyond task-specific knowledge, an ideal CPT process also requires preserving the model's foundational, general-purpose abilities. We evaluated the models on general benchmarks to measure this effect. The results, presented in Table 2, highlight another critical advantage of RFT.

**SFT harms general capabilities in both CL and MTL.** Our experiments reveal that SFT causes significant *base model degradation* (Liu et al., 2025a). SFT induces a severe performance drop of ↓**16.9%** on the challenging MMLU-Pro benchmark (47.5% → 30.6%). Crucially, this is not merely an artifact of sequential learning; even multi-task SFT (MTL (SFT)), which trains on all data simultaneously, still causes a severe drop of ↓**14.3%** on the same benchmark. A similar trend is evident on MMMU, where SFT and MTL (SFT) cause performance to decline by ↓**12.0%** and ↓**4.3%** respectively. This demonstrates that the SFT paradigm itself appears harmful to the model's foundational capabilities.

**RFT preserves and enhances general capabilities.** In contrast to the capability decay observed across all SFT methods, the RFT paradigm effectively safeguards the model's general abilities. GRPO, in particular, often *enhances* these abilities. For instance, GRPO improves performance on MMMU by ↑**2.1%** (52.1% → 54.2%). Crucially, RFT also improves model general capabilities, with GRPO improving the POPE score by ↑**1.9%** (86.6% → 88.5%) and reducing the tendency for hallucination. This clear difference highlights that RFT is a more robust paradigm for continual post-training.

## 5 HOW DOES RFT MITIGATE FORGETTING?

To investigate the mechanisms behind RFT's remarkable stability, this section presents a series of ablation studies based on the popular and representative GRPO algorithm Shao et al. (2024).

### 5.1 THE ROLES OF CoT AND KL PENALTY

We test two primary hypotheses: **(1)** The KL-divergence penalty, by regularizing policy updates, acts as a form of knowledge distillation (Li & Hoiem, 2017) that preserves past knowledge. **(2)** The complex reasoning

Table 4: General capabilities evaluation for ablation models. Each benchmark is evaluated with and without CoT prompts to provide a comprehensive view.

| Benchmark | Eval CoT | Base | GRPO | GRPO w/o CoT | GRPO w/o KL |
|-----------|----------|------|------|--------------|-------------|
| MMMU | ✔ | 52.1 | 54.2 (↑**2.1**) | 51.8 (↓**0.3**) | 52.2 (↑**0.1**) |
| | ✗ | 48.7 | 51.0 (↑**2.3**) | 51.6 (↑**2.9**) | 49.2 (↑**0.5**) |
| MMLU-Pro | ✔ | 47.5 | 48.3 (↑**0.8**) | 48.9 (↑**1.4**) | 45.2 (↓**2.3**) |
| | ✗ | 40.0 | 42.2 (↑**2.2**) | 41.9 (↑**1.9**) | 42.3 (↑**2.3**) |
| POPE | ✔ | 86.6 | 88.5 (↑**1.9**) | 85.3 (↓**1.3**) | 74.2 (↓**12.4**) |
| | ✗ | 82.2 | 84.0 (↑**1.8**) | 88.7 (↑**6.5**) | 87.6 (↑**5.4**) |

structure of CoT builds more abstract and resilient knowledge representations, protecting them from being overwritten. Thus, we evaluate three GRPO variants against the SFT baseline: *GRPO w/o KL*: trained with CoT prompts but without the KL penalty term. *GRPO w/o CoT*: trained without CoT prompts, using direct question-answering format but retaining the KL penalty.

**KL penalty is not the primary factor for preserving task-specific knowledge.** As shown in Table 3, removing the KL penalty (*GRPO w/o KL*) causes no degradation in performance on the continual learning sequence. The final average accuracy remains, demonstrating that the KL penalty is *not* the primary mechanism preventing task-specific catastrophic forgetting. However, it is crucial to note that the training process without the KL penalty exhibits significant instability in the later stages of the task sequence. These results are obtained after multiple attempts, re-initializing from the previous task's checkpoint to achieve a convergent outcome, which suggests KL penalty plays a critical role in stabilizing the RFT process.

**CoT is a performance booster, not a forgetting mitigator.** Our second hypothesis is also not supported by the data. The model trained without CoT (*GRPO w/o CoT*) still strongly resists forgetting, maintaining a high average accuracy across the task sequence (Table 3). In fact, it outperforms GRPO on VizWiz ( 63.8% vs. 51.8%). The general capabilities evaluation in Table 4 further confirms this conclusion. The *GRPO w/o CoT* model remains robust, and it achieves the highest score on the POPE benchmark (**88.7%**) when tested in non-CoT format evaluation. This demonstrates that while CoT can enhance performance on certain types of tasks, it is not the mechanism responsible for RFT's resistance to catastrophic forgetting. Besides, as shown in Table 3, we observe that for *GRPO w/o KL*, using CoT during inference would lead to notable hallucination.

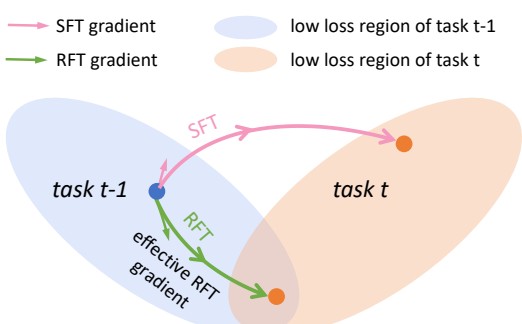

Figure 4: Only samples producing non-zero reward variance yield effective policy gradients; RIF-RFT improves sample efficiency by focusing training on such informative samples.

## 5.2 IMPLICIT REGULARIZATION FROM REWARD VARIANCE

To build intuition for the empirical resilience of RFT, we analyze its gradient dynamics in the context of continual learning. Our analysis suggests that RFT's forgetting mitigation stems from an *implicit regularization* mechanism, where the learning signal itself modulates the update strength. To explore this intuition, we adopt the concept of *forgetting risk* from continual learning theory (Kirkpatrick et al., 2017), using the Fisher Information Matrix (FIM) as a tool to quantify parameter sensitivity to past tasks. This allows us to conceptually link the structure of RFT's gradients to knowledge retention.

**Definition 5.1** (Forgetting Risk). Let $\mathcal{D}_{1:k-1}$ be the data from all previously learned tasks. The FIM is defined as $F_{k-1} \triangleq \mathbb{E}_{(x,a^*)\sim\mathcal{D}_{1:k-1}}[\nabla_\theta \log \pi_\theta(a^*|x)(\nabla_\theta \log \pi_\theta(a^*|x))^\top]$. The **_forgetting risk_** of a gradient update $g$ for the current task $k$ is defined as its squared Mahalanobis norm with respect to the FIM of past tasks:

$$\mathcal{R}(g) \triangleq g^\top F_{k-1} g. \tag{9}$$

This risk measures the update's magnitude in parameter subspaces critical for prior knowledge. Note that $F_{k-1}$ is a theoretical construct for our analysis and is not computed in practice.

For a single data point $(x_k, a_k^*) \in \mathcal{D}_k$, the SFT loss gradient is $g_{\text{SFT}} = -\nabla_\theta \log \pi_\theta(a_k^*|x_k)$. In contrast, the RFT policy gradient for a sampled response $a \sim \pi_\theta(\cdot|x_k)$ is $g_{\text{RFT}}(a) = A(x_k,a)\nabla_\theta \log \pi_\theta(a|x_k)$, where $A(x_k,a)$ is an advantage function ( $r(x_k,a) - b(x_k)$ ).

Table 5: Performance and data efficiency comparison of our proposed RIF-RFT.

| Method | SciQA | TextVQA | VizWiz | GQA | Geo. | PathVQA | sCLEVR | AvgAcc | FM |
|--------|-------|---------|--------|-----|------|---------|--------|--------|-----|
| SFT | 76.1 | 55.8 | 46.8 | 58.5 | 20.2 | 62.2 | 58.2 | 54.0 | -10.4 |
| GRPO | 93.0 | 74.8 | 51.8 | 65.9 | 38.4 | 41.3 | 54.2 | 60.0 | -2.3 |
| RIF-RFT | 92.9 | 73.7 | 46.6 | 63.0 | 32.3 | 40.5 | 53.7 | 57.5 | -4.5 |
| Data Kept | 81.4% | 45.6% | 42.1% | 67.6% | 37.2% | 42.5% | 52.3% | - | - |

The following proposition establishes a conceptual link between the expected forgetting risk of an RFT update and that of an SFT update, highlighting the central role of reward variance.

**Proposition 5.2** (RFT's Implicit Regularization Effect). *Consider a single update on task $k$ at parameters $\theta_{k-1}$. Let the rewards be normalized, $r(x_k, a) \in [0, 1]$. Under the technical assumptions specified in Appendix B, the expected forgetting risk of an RFT update is related to the SFT risk by:*

$$\mathbb{E}_{a \sim \pi_{\theta_{k-1}}}[\mathcal{R}(g_{RFT}(a))] \approx Var_{a \sim \pi_{\theta_{k-1}}}[r(x_k, a)] \cdot \mathcal{R}(g_{SFT}), \tag{10}$$

*where the approximation holds when an error term $\mathcal{E}$, capturing second-order effects, is small. The term $Var[r(x_k, a)]$ is bounded by $1/4$ for normalized rewards.*

The full proof is provided in Appendix B. Proposition 5.2 offers an intuition: the expected impact of an RFT update on prior knowledge is not fixed but is dynamically scaled by the reward variance. For an uncertain sample where the model generates diverse responses with high reward variance, the update magnitude in sensitive directions is naturally dampened, thus protecting established knowledge. Conversely, for samples where the model produces consistently high-reward responses, the update is more aggressive. This inherent, data-dependent regularization mechanism contrasts with SFT's uniform, high-variance gradients, offering a compelling explanation for the stability observed in our experiments and illustrated in Figure 4.

### 5.3 RIF-RFT: ENHANCING STABILITY AND EFFICIENCY OF RFT

Our analysis in Section 5.2 reveals that RFT's resilience to forgetting is based in a reward-variance-scaled regularization. However, this mechanism's effectiveness relies on the model's ability to generate responses that produce a meaningful reward signal. We identify a critical failure mode when the model is faced with incompetent samples: training instances for which the current policy $\pi_\theta$ consistently fails to produce non-zero rewarded outputs. For such samples, the advantage estimates $A(x, a)$ collapse to zero or are dominated by noise, yielding no effective policy gradient. This reduces sample efficiency without contributing to meaningful learning.

To address this challenge, we propose a simple yet effective method: **R**ollout-based **I**nstance **F**iltering for **RFT** (RIF-RFT). The motivation, illustrated in Figure 4, is to prune the training data by identifying and discarding these incompetent samples before the RFT training. By filtering them out, RFT focuses its capacity on instances where it can receive a productive learning signal, stabilizing the regularization effect and improving efficiency. Note that as training progresses, samples that initially yielded zero reward may become learnable. RIF-RFT trades this marginal adaptability for computational savings.

The mechanism is formalized in Algorithm 1 in Appendix C. For each instance in a new task's dataset $\mathcal{D}_k$, we perform a small number of policy rollouts. If at least one of these rollouts produces a response with a reward greater than a minimal threshold $\tau$, we classify the instance and retain it in $\mathcal{D}_k^{\text{filt}}$. As shown in Table 5, while full-data GRPO achieves the best performance, it processes many samples with zero reward variance that yield no effective policy gradients. RIF-RFT addresses this inefficiency by filtering such samples a priori, maintaining strong anti-forgetting properties. This demonstrates a compelling trade-off between efficiency and robustness.

## 6 CONCLUSION

This work presents a comprehensive investigation into the role of the fundamental learning paradigm in continual post-training for MLLMs. Our central finding is that RFT naturally mitigates the catastrophic forgetting that plagues the standard SFT. Through extensive experiments, we demonstrate that while SFT leads to severe degradation of both previously learned task-specific skills and general capabilities, RFT paradigms inherently preserve those knowledge, achieving performance comparable to an offline multi-task learning setting. Our analysis suggests this superiority stems not from explicit mechanisms like CoT or KL regularization, but from an implicit regularization effect inherent to RFT. We provide a theoretical perspective that attributes this stability to reward-variance-scaled updates, which naturally protect previously acquired knowledge by moderating

learning on uncertain samples. Finally, we introduce RIF-RFT, an efficient instance filtering method that improves the stability and sample efficiency of RFT without compromising its robustness. This research suggests that RFT is not merely an alternative but a fundamentally more suitable paradigm for the continual and lifelong adaptation of foundation models.

## ETHICS STATEMENT

This research focuses on the fundamental learning paradigms for continual post-training of Multimodal Large Language Models. All experiments were conducted on publicly available and well-established academic benchmarks. Our work did not involve human subjects, private data, or generation of personally identifiable information.

## REPRODUCIBILITY STATEMENT

To ensure the reproducibility of our findings, we provide comprehensive details throughout the paper and in the appendix. The source code is available in the supplementary material. Our experiments are based on the publicly available Qwen2.5-VL-7B-Instruct model. All implementation details are documented in Section 4 and the Appendix. We use standard public datasets, with detailed descriptions provided in Section 4 and the Appendix.

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

# A DATASET INFORMATION

```
NoCoT Prompt Template

Question:
<image>What is the probability that a Nile tilapia fish...
A. 2/4   B. 3/4 ... E. 4/4
You MUST provide the final answer directly.

Answer:
E
```

```
CoT Prompt Template

Question:
<image>What is the probability that a Nile tilapia fish...
A. 2/4   B. 3/4 ... E. 4/4
You FIRST think about the reasoning process...
The reasoning process MUST BE enclosed within
<think> </think> tags.
The final answer MUST BE put in \boxed{}.

Answer:
<think>
The Punnett square shows...
Therefore, the probability is 4/4. The correct answer is E.
</think>
\boxed{E}
```

Figure 5: Example prompt templates w/o and w/ CoT.

**Multimodal Datasets for Continual Post-Training and Evaluation.** Our study utilized a diverse suite of vision-language datasets for both model training and comprehensive evaluation of various multimodal capabilities, along with specialized benchmarks to assess knowledge retention and nuanced multimodal challenges. Here is a brief introduction to these datasets:

*Multimodal Datasets for Continual Post-Training:*

- **ScienceQA** (Saikh et al., 2022) presents multimodal science questions requiring complex reasoning over diagrams, text, and general knowledge.
- **TextVQA** (Singh et al., 2019) focuses on questions that necessitate reading and inferring from text embedded within images.
- **VizWiz** (Gurari et al., 2018) comprises real-world image-based questions posed by visually impaired individuals, often involving ambiguity.
- **GQA** (Hudson & Manning, 2019) is designed for compositional question answering over real-world images with a strong emphasis on spatial understanding and object relationships.
- **Geometry3K** (Lu et al., 2021): This subset of MathVista (Lu et al., 2024) comprises multi-choice geometry problems equipped with dense annotations in formal language for both diagrams and text, specifically designed to evaluate complex geometric reasoning skills.
- **PathVQA** (He et al., 2020) provides medical visual question answering on pathology images that demand specialized domain knowledge.
- **Super-CLEVR** (Li et al., 2023b) is a synthetic dataset crafted to rigorously test complex relational and logical reasoning.

*Benchmarks for General Knowledge Evaluation:*

- **MMMU** (Yue et al., 2024) is comprehensive benchmark comprising 11.5K college-level, multi-discipline multimodal tasks with diverse image types, demanding deliberate reasoning.

- **MMLU-Pro** (Wang et al., 2024b) is an enhanced benchmark designed for more discriminative evaluation of large language models, featuring more challenging and reasoning-focused questions with ten multiple-choice options, sourced from various academic and STEM fields.

- **POPE** (Li et al., 2023a) is a benchmark introduced to systematically investigate and assess object hallucination in vision-language large models through an improved polling-based query method.

## B  PROOF AND TECHNICAL DETAILS FOR PROPOSITION 5.2

We provide the detailed derivation for Proposition 5.2, which establishes the relationship between the forgetting risks of RFT and SFT.

**Proposition 5.2.** *Let the rewards be normalized, $r(x_k, a) \in [0, 1]$. Under Assumption B.2, the expected forgetting risk of an RFT update is related to the SFT risk by:*

$$\mathbb{E}_{a \sim \pi_{\theta_{k-1}}}[\mathcal{R}(g_{RFT}(a))] = Var_{a \sim \pi_{\theta_{k-1}}}[r(x_k, a)] \cdot \mathcal{R}(g_{SFT}) + \mathcal{E}$$

*where $\mathcal{E}$ is an error term characterized in the proof.*

**Definition B.1** (Importance-Weighted Score Norm (IWSN)). For a response $a$, we define its IWSN as the squared norm of its score function, weighted by the FIM of past tasks:

$$I(a) \triangleq (\nabla_\theta \log \pi_\theta(a|x_k))^\top F_{k-1} (\nabla_\theta \log \pi_\theta(a|x_k))$$

**Assumption B.2** (Technical Assumptions). *Our analysis relies on the following two technical assumptions for a given data point $x_k$ and parameters $\theta_{k-1}$:*

1. ***Bounded Covariance.*** *The covariance between the squared advantage and the IWSN is bounded: $Cov(A(a)^2, I(a)) = \epsilon_1$, where $\epsilon_1$ is a small error term. This implies that the magnitude of an advantage signal is not strongly correlated with the gradient's impact on prior tasks.*

2. ***Centered Policy Expectation.*** *The expected IWSN under the current policy is close to the IWSN of the ground-truth response: $\mathbb{E}_{a \sim \pi_{\theta_{k-1}}}[I(a)] - I(a_k^*) = \delta$, where $\delta$ is another small error term. This holds when the policy $\pi_{\theta_{k-1}}$ generates responses that, on average, have a similar gradient geometry to the ground-truth response with respect to past tasks.*

**Lemma B.3** (Variance of Advantage). *For policy gradient methods, using the reward baseline $b(x_k) = \mathbb{E}_{a \sim \pi_\theta}[r(x_k, a)]$ minimizes the variance of the gradient estimator. With this optimal baseline, the expected squared advantage equals the reward variance:*

$$\mathbb{E}_{a \sim \pi_\theta}[A(x_k, a)^2] = Var_{a \sim \pi_\theta}[r(x_k, a)]$$

*Proof.* By definition, $A(x_k, a) = r(x_k, a) - b(x_k)$. We have:

$$\mathbb{E}[A^2] = \mathbb{E}[(r - b)^2] = \mathbb{E}[r^2] - 2b\mathbb{E}[r] + b^2$$

Since $b = \mathbb{E}[r]$, this simplifies to $\mathbb{E}[r^2] - 2(\mathbb{E}[r])^2 + (\mathbb{E}[r])^2 = \mathbb{E}[r^2] - (\mathbb{E}[r])^2 = \text{Var}[r]$  □

*Proof of Proposition 5.2.* Let us analyze the forgetting risks at parameters $\theta = \theta_{k-1}$ for a single data point $(x_k, a_k^*)$.

The SFT loss gradient is $g_{\text{SFT}} = -\nabla_\theta \log \pi_\theta(a_k^*|x_k)$. Its forgetting risk is deterministic:

$$\mathcal{R}(g_{\text{SFT}}) = g_{\text{SFT}}^\top F_{k-1} g_{\text{SFT}} = (\nabla_\theta \log \pi_\theta(a_k^*|x_k))^\top F_{k-1} (\nabla_\theta \log \pi_\theta(a_k^*|x_k)) = I(a_k^*)$$

The RFT gradient for a sampled response $a \sim \pi_\theta(\cdot|x_k)$ is $g_{\text{RFT}}(a) = A(x_k, a)\nabla_\theta \log \pi_\theta(a|x_k)$. We compute the expectation of its forgetting risk:

$$
\begin{aligned}
\mathbb{E}[\mathcal{R}(g_{\text{RFT}})] &= \mathbb{E}_{a \sim \pi_\theta}\left[(g_{\text{RFT}}(a))^\top F_{k-1}(g_{\text{RFT}}(a))\right] \\
&= \mathbb{E}_{a \sim \pi_\theta}\left[A(x_k, a)^2 \cdot (\nabla_\theta \log \pi_\theta(a|x_k))^\top F_{k-1}(\nabla_\theta \log \pi_\theta(a|x_k))\right] \\
&= \mathbb{E}_{a \sim \pi_\theta}\left[A(x_k, a)^2 \cdot I(a)\right] \quad (11)
\end{aligned}
$$

We use the covariance identity $\mathbb{E}[XY] = \mathbb{E}[X]\mathbb{E}[Y] + \mathrm{Cov}(X,Y)$ to decompose Eq. 11:

$$
\begin{aligned}
\mathbb{E}[\mathcal{R}(g_{\mathrm{RFT}})] &= \mathbb{E}[A(a)^2] \cdot \mathbb{E}[I(a)] + \mathrm{Cov}(A(a)^2, I(a)) \\
&= \mathrm{Var}[r(x_k,a)] \cdot \mathbb{E}[I(a)] + \epsilon_1 \quad \text{(from Lemma B.3 and Assumption B.2.1)} \\
&= \mathrm{Var}[r(x_k,a)] \cdot (I(a_k^*) + \delta) + \epsilon_1 \quad \text{(from Assumption B.2.2)} \\
&= \mathrm{Var}[r(x_k,a)] \cdot I(a_k^*) + \mathrm{Var}[r(x_k,a)]\delta + \epsilon_1 \\
&= \mathrm{Var}[r(x_k,a)] \cdot \mathcal{R}(g_{\mathrm{SFT}}) + \underbrace{\mathrm{Var}[r(x_k,a)]\delta + \epsilon_1}_{\mathcal{E}}
\end{aligned}
$$

This completes the proof. The error term $\mathcal{E} = \mathrm{Var}[r(x_k,a)]\delta + \epsilon_1$ is small under reasonable conditions. Specifically, $\delta$ is small when the current policy is not drastically different from one that produces the ground-truth response, a condition met after initial task adaptation. $\epsilon_1$ is small if there is no systematic correlation between a response's quality (reflected in its advantage) and its gradient's impact on prior tasks, which is a mild assumption for complex, high-dimensional models. While this analysis provides an approximation rather than a strict bound, it formalizes the core intuition that reward variance acts as a natural, implicit regularizer, offering a strong theoretical motivation for the empirical stability of RFT in continual post-training. $\square$

## C    PSEUDO CODE OF RIF-RFT

---

**Algorithm 1: R**ollout-based **I**nstance **F**iltering for **RFT** (`RIF-RFT`)

---

**Input:** New task training set: $\mathcal{D}_k = \{(x_i, y_i^*)\}_{i=1}^M$; Current model policy: $\pi_\theta$; Number of
       rollouts per input: $N$; Reward threshold: $\tau$
**Initialize** filtered dataset: $\mathcal{D}_k^{\mathrm{filt}} \leftarrow \emptyset$;
**for** *each sample* $(x_i, y_i^*) \in \mathcal{D}_k$ **do**
    Initialize $R_{\mathrm{sum}} \leftarrow 0$;
    **for** $j = 1$ **to** $N$ **do**
        Sample a response: $y_{ij} \sim \pi_\theta(\cdot \mid x_i)$;
        Compute reward: $R(y_{ij})$;
        Update: $R_{\mathrm{sum}} \leftarrow R_{\mathrm{sum}} + R(y_{ij})$;
    **if** $R_{sum}/N > \tau$ **then**
        Add $(x_i, y_i^*)$ to $\mathcal{D}_k^{\mathrm{filt}}$;

Perform standard RFT on the filtered dataset $\mathcal{D}_k^{\mathrm{filt}}$ to obtain $\pi_{\theta'}$;
**Output:** Updated model $\pi_{\theta'}$

---

## D    ROBUSTNESS AND EFFICIENCY ANALYSIS

To validate the generality of our findings beyond the primary Qwen2.5-VL-7B-Instruct model, we conduct extensive experiments across different architectures, model scales, and task domains. These additional studies ensure that the observed forgetting mitigation is an intrinsic property of the RFT paradigm rather than an artifact of a specific model configuration.

### D.1    GENERALIZATION ACROSS ARCHITECTURES AND SCALES

#### D.1.1    TEXT-ONLY TASKS

We first evaluate whether RFT's forgetting mitigation extends to text-only domains. We utilize the text-only Qwen2.5-7B-Instruct (Yang et al., 2024) and evaluate it on two diverse benchmarks: GSM8K (Cobbe et al., 2021) for mathematical reasoning and USMLE (Jin et al., 2020) for medical knowledge. These tasks provide clear correctness signals suitable for both SFT and RFT paradigms. As shown in Table 6, RFT consistently outperforms SFT. For instance, in the GSM8K → USMLE sequence, RFT maintains a forgetting measure (FM) of -1.8%, whereas SFT suffers a significant drop with an FM of -10.4%.

#### D.1.2    MODEL SCALE ANALYSIS

We further evaluate the impact of model scale on forgetting mitigation using Qwen2.5-VL-3B-Instruct (Bai et al., 2025a) and the larger Qwen3-VL-8B-Instruct (Yang et al., 2025) on a subset of our benchmark tasks

Table 6: Performance evaluation on text-only tasks using Qwen2.5-7B-Instruct.

| Method | Task Order | GSM8K | USMLE | AvgAcc | FM |
|--------|-----------|-------|-------|--------|-----|
| GRPO | GSM8K→USMLE | 84.2 | 62.3 | 73.3 | -1.8 |
|  | USMLE→GSM8K | 85.1 | 60.7 | 72.9 | -1.2 |
| SFT | GSM8K→USMLE | 71.3 | 58.2 | 64.8 | -10.4 |
|  | USMLE→GSM8K | 82.4 | 49.6 | 66.0 | -8.7 |

(sCLEVR, ScienceQA, and TextVQA). The results are summarized in Table 7. We observe that RFT maintains near-zero forgetting across both scales . Notably, the larger 8B model exhibits stronger resilience to catastrophic forgetting under the RFT paradigm compared to the 3B model.

Table 7: Performance comparison across different model scales.

| Model Size | Method | sCLEVR | SciQA | TextVQA | AvgAcc | FM |
|-----------|--------|--------|-------|---------|--------|-----|
| 3B | GRPO | 57.8 | 92.7 | 72.8 | 74.4 | -0.4 |
|  | SFT | 51.5 | 92.3 | 67.6 | 70.5 | -4.4 |
| 8B | GRPO | 57.0 | 96.3 | 76.1 | 76.5 | -0.2 |
|  | SFT | 48.2 | 91.5 | 68.9 | 69.5 | -7.1 |

## D.2 COMPARISON WITH CL METHODS

To compare RFT's performance against established CL techniques, we compare it with Experience Replay (ER) (Schaul et al., 2015), widely considered one of the most effective baselines. We implement ER with a 25% replay ratio, which represents the upper range suggested by recent work on LLM continual post-training (Abbes et al., 2025). As detailed in Table 8, while ER improves upon vanilla SFT (FM improves from -4.4% to -2.8%), it still lags behind RFT (-0.4%). Furthermore, ER introduces significant storage overhead and potential negative transfer, whereas RFT achieves superior stability inherently without requiring external memory buffers.

Table 8: Comparison between RFT, SFT, and SFT with ER on Qwen2.5-VL-3B-Instruct.

| Method | sCLEVR | SciQA | TextVQA | AvgAcc | FM |
|--------|--------|-------|---------|--------|-----|
| SFT | 51.5 | 92.3 | 67.6 | 70.5 | -4.4 |
| SFT + ER (25%) | 53.2 | 92.1 | 64.5 | 69.9 | -2.8 |
| GRPO | 57.8 | 92.7 | 72.8 | **74.4** | **-0.4** |

## D.3 ROBUSTNESS TO TASK ORDERING

Continual learning performance is often sensitive to the task order. We evaluate RFT's robustness by testing two distinct task orderings on both 3B and 8B models. The results in Table 9 show that the Forgetting Measure remains consistently low (ranging from -0.2% to -0.4%) regardless of the order.

## D.4 COMPUTATIONAL EFFICIENCY OF RIF-RFT

Regarding the computational overhead of our proposed RIF-RFT method, we provide a detailed efficiency analysis in Table 10 on 8×H800 GPUs. The RIF-RFT process consists of a filtering phase (inference only) followed by training on the filtered data. Our analysis reveals that the filtering overhead is negligible (<2% of total time) because it avoids the costly backpropagation step. Crucially, by reducing the dataset size for the subsequent training phase, RIF-RFT achieves a ∼44% reduction in total wall-clock time compared to standard GRPO, demonstrating that our method improves both computational and sample efficiency.

## D.5 ABLATION ON FILTERING THRESHOLD IN RIF-RFT

In RIF-RFT, the filtering threshold $\tau$ determines which samples are retained for training. We use $\tau = 0$ as default, meaning samples are retained if they achieve any non-zero reward across rollouts. The threshold $\tau$

Table 9: Performance evaluation under different task orderings.

| Model | Task Order | Task 1 | Task 2 | Task 3 | AvgAcc | FM |
|---|---|---|---|---|---|---|
| Qwen2.5-VL-3B | sCLEVR→SciQA→TextVQA | 57.8 | 92.7 | 72.8 | 74.4 | -0.4 |
| | TextVQA→SciQA→sCLEVR | 72.8 | 92.1 | 57.8 | 74.2 | -0.3 |
| Qwen3-VL-8B | sCLEVR→SciQA→TextVQA | 57.0 | 96.3 | 76.1 | 76.5 | -0.2 |
| | TextVQA→SciQA→sCLEVR | 76.1 | 96.8 | 55.6 | 76.2 | -0.4 |

Table 10: Wall-clock time (hours) analysis comparing standard GRPO and RIF-RFT.

| Dataset | GRPO | RIF-RFT (Train) | RIF-RFT (Filter) | RIF-RFT |
|---|---|---|---|---|
| ScienceQA | 6.4 | 5.2 | 0.13 | 5.33 |
| TextVQA | 30.9 | 13.8 | 0.31 | 14.11 |
| VizWiz | 19.5 | 8.0 | 0.20 | 8.20 |
| GQA | 72.6 | 48.6 | 0.50 | 49.10 |
| Geometry3K | 2.3 | 0.6 | 0.02 | 0.62 |
| PathVQA | 15.4 | 5.6 | 0.15 | 5.75 |
| sCLEVR | 6.7 | 3.4 | 0.11 | 3.51 |
| Total | 153.8 | 85.2 | 1.42 | **86.62** |

controls a trade-off between data quality and quantity: higher thresholds retain only samples where the model succeeds more consistently, but this reduces the volume of training data. We conduct ablation experiments on the task sequence sCLEVR → ScienceQA → TextVQA using Qwen2.5-VL-3B. Results are presented in Table 11.

Table 11: Ablation on filtering threshold $\tau$.

| $\tau$ | sCLEVR | SciQA | TextVQA | AvgAcc |
|---|---|---|---|---|
| 0 | 57.2 | 92.5 | 72.1 | 73.9 |
| 0.1 | 56.8 | 92.1 | 71.4 | 73.4 |
| 0.2 | 55.9 | 91.6 | 70.2 | 72.6 |

Our default setting achieves the highest overall performance. This suggests that samples where the model has low but non-zero reward provide effective gradient signals for policy improvement. As $\tau$ increases, performance degrades across all tasks due to reduced training data volume. We recommend $\tau = 0$, which maximizes the retention of informative training instances.

# E STATEMENT ON THE USE OF LLM

We disclose that LLMs were used to assist with writing. Their role was limited to improving the grammar and overall readability of the manuscript. The core research ideas, experimental results, and scientific claims presented are the work of the authors.

