# OpenReview forum: "Reinforcement Fine-Tuning Naturally Mitigates Forgetting in Continual Post-Training"
_ICLR.cc/2026/Conference — ICLR 2026 Conference Desk Rejected Submission_

### Official Review · Reviewer_wgFr · 2025-10-22

**Soundness:** 3
**Presentation:** 3
**Contribution:** 3
**Rating:** 4
**Confidence:** 4

**Summary:**

This paper focuses on the catastrophic forgetting problem in the continual post-training (CPT) of multimodal large language models (MLLMs) and conducts a systematic comparison of the knowledge retention capabilities between the two paradigms: supervised fine-tuning (SFT) and reinforcement fine-tuning (RFT). Through extensive experiments, it verifies that RFT has an inherent forgetting mitigation advantage, reveals the implicit regularization mechanism driven by reward variance, and finally proposes the Rollout-based Instance Filtering for RFT (RIF-RFT) algorithm to enhance the stability and efficiency of RFT.

**Strengths:**

1. This paper reveals the inherent forgetting mitigation property in the reinforcement fine-tuning (RFT) paradigm, which endows RFT with superiority in post-training.
2. The mechanism analysis of forgetting mitigation under RFT is in-depth and supported by theories.

**Weaknesses:**

1. Experiments are only conducted based on Qwen2.5-VL-7B-Instruct, making it impossible to confirm whether the forgetting mitigation advantage of RFT is independent of the architecture.
2. There is a lack of direct comparisons with mainstream continual learning methods in terms of knowledge retention performance, storage overhead, and computational efficiency.
3. The selection method of the filtering threshold in RIF-RFT is not provided, and relevant ablation experiments are lacking.

**Questions:**

1. There is a correlation between continual learning performance and task order; can the model's forgetting resistance effect be verified under different task orders?
2. Only models of the 7B scale are verified in the paper; can the impact of model parameter scale on the forgetting mitigation effect of RFT be explored?
3. The experiment focuses on multimodal models: what is the degree of protection provided by RFT for the knowledge of different modalities, and is there any preference?

---

> ### Author Response · Authors · 2025-11-24
> **Response to Reviewer wgFr (1/2)**
>
> We sincerely thank the reviewer for the constructive feedback. We have conducted additional experiments to address your concerns. Below, we have provided detailed responses to each point as follows. Please feel free to let us know if you have any additional concerns or questions.
>
> **W1 & Q2:**  Generalizability across architectures and model scales
>
> We appreciate the reviewer's concern about the generalizability. While our main experiments focused on Qwen2.5-VL-7B-Instruct, we acknowledge this limitation and have conducted additional experiments to verify that our findings generalize across architectures and model sizes. We have incorporated these experimental results into Appendix D.1 of the revised paper.
>
> - We would like to clarify that we chose Qwen2.5‑VL‑7B‑Instruct as the base model because it has become **one of the de‑facto standard backbones** for R1‑style RFT in the open‑source community. Recent multimodal reinforcement fine tuning works[1-5] **all adopt Qwen2.5‑VL‑Instruct** as main backbone. Studying continual post‑training on this backbone therefore makes our findings directly comparable to these works. Moreover, among the existing studies that explicitly compare SFT and RFT on MLLMs [2] [6], Qwen2.5‑VL is also dominant choice.
> - Nevertheless, we agree that broader validation strengthens our claims, and we have conducted additional experiments on text-only tasks and more other large vision-language models (e.g., Qwen3-VL) as follows.
> - First, to validate generality beyond vision-language models, we conducted experiments on text-only tasks using Qwen2.5-7B-Instruct [7] on GSM8K  (mathematical reasoning) [8] and USMLE (medical knowledge) [9]. These two tasks represent different knowledge types and are standard benchmarks where both SFT and RFT can be applied with clear correctness signals.
>
> **Table 1**: Results on Qwen2.5-7B-Instruct. RFT results shown in our tables are obtained using GRPO.
>
> | Method | Task Order  | GSM8K | USMLE | FM    |
> | ------ | ----------- | ----- | ----- | ----- |
> | RFT    | GSM8K→USMLE | 84.2  | 62.3  | -1.8  |
> |        | USMLE→GSM8K | 85.1  | 60.7  | -1.2  |
> | SFT    | GSM8K→USMLE | 71.3  | 58.2  | -10.4 |
> |        | USMLE→GSM8K | 82.4  | 49.6  | -8.7  |
>
> Table 1 confirms that RFT's forgetting mitigation extends to text domains, demonstrating architecture-independent benefits.
>
> -  Second, we also evaluated both smaller (3B) and larger (8B) models using Qwen2.5-VL-3B-Instruct [10] and Qwen3-VL-8B-Instruct [11] on 3 tasks:
>
> **Table 2**: Performance across different model scales (Task Order: sCLEVR→ScienceQA→TextVQA). We report performance after each task.
>
> | Model Size | Method |   Stage   | sCLEVR | ScienceQA | TextVQA |  FM  |
> | :--------: | :----: | :-------: | :----: | :-------: | :-----: | :--: |
> |     3B     |        |   Base    |  43.8  |   31.0    |  68.6   |  -   |
> |            |  RFT   |  sCLEVR   |  58.7  |     -     |    -    |  -   |
> |            |        | ScienceQA |  57.9  |   92.6    |    -    |  -   |
> |            |        |  TextVQA  |  57.8  |   92.7    |  72.8   | -0.4 |
> |            |  SFT   |  sCLEVR   |  58.5  |     -     |    -    |  -   |
> |            |        | ScienceQA |  41.3  |   94.1    |    -    |  -   |
> |            |        |  TextVQA  |  51.5  |   92.3    |  67.6   | -4.4 |
> |     8B     |        |   Base    |  17.6  |   44.9    |  71.2   |  -   |
> |            |  RFT   |  sCLEVR   |  57.3  |     -     |    -    |  -   |
> |            |        | ScienceQA |  57.2  |   96.3    |    -    |  -   |
> |            |        |  TextVQA  |  57.0  |   96.3    |  76.1   | -0.2 |
> |            |  SFT   |  sCLEVR   |  58.1  |     -     |    -    |  -   |
> |            |        | ScienceQA |  45.2  |   95.8    |    -    |  -   |
> |            |        |  TextVQA  |  48.2  |   91.5    |  68.9   | -7.1 |
>
> * Table2 demonstrates that RFT consistently achieves near-zero forgetting across scales, whereas SFT shows varying degrees of catastrophic forgetting.
> * Interestingly, we observe that Qwen3-VL-8B shows even stronger forgetting resistance than Qwen2.5-VL-3B model, suggesting larger model size further enhance RFT's benefits.
> * We observe that the 8B model has lower zero-shot performance on sCLEVR compared to the 3B model, likely due to sensitivity to the synthetic data distribution. However, RFT successfully enables it to learn the task reaching 57% and subsequently protects this knowledge. This shows that RFT protects acquired knowledge effectively, regardless of the starting point.

---

> ### Author Response · Authors · 2025-11-24
> **Response to Reviewer wgFr (2/2)**
>
> **W2:** Comparison with mainstream CL methods
>
> * We conducted direct comparisons with experience replay (ER), which is one of the most effective method in classical CL. At each stage, we sample a fixed proportion of examples from every seen task and mix them with the current task’s data. Recent work [12] [13] on LLM continual pre‑training and fine‑tuning suggests that on the order of 5–25% of tokens are sufficient to recover most of the forgetting reduction achievable with ER. We therefore use SFT with a 25% replay ratio, which we treat as the upper end of this range. Note that, under this setting, the memory buffer requirement grows linearly with the number of tasks. This comparison has been added to Appendix D.2 of the revised paper.
>
> **Table 3:** RFT vs. SFT with ER on Qwen2.5-VL-3B. We compare SFT, SFT with ER, and RFT approach across three continual learning tasks.
>
> | Method  | sCLEVR | ScienceQA | TextVQA |  FM  |
> | :-----: | :----: | :-------: | :-----: | :--: |
> |   SFT   |  51.5  |   92.3    |  67.6   | -4.4 |
> | SFT+ ER |  53.2  |   92.1    |  64.5   | -2.8 |
> |   RFT   |  57.8  |   92.7    |  72.8   | -0.4 |
>
> * Table 3 shows that SFT with ER still exhibits noticeable forgetting. In contrast, RFT achieves near-zero forgetting.
> * SFT+ER shows a slight performance drop on the current task, likely due to **negative transfer** from the replay buffer. RFT avoids this trade-off, achieving both high plasticity and stability.
>
>
> **W3:** Selection of filtering threshold in RIF-RFT
>
> We appreciate the reviewer's attention to this implementation detail. In RIF-RFT, we use $\tau = 0$ in our  experiments, meaning samples are retained if they achieve any non-zero reward across rollouts. We conduct ablation studies with different thresholds to investigate the trade-off between data quality and quantity.
>
> **Table 4:** Ablation on filtering threshold $\tau$ (sCLEVR → ScienceQA → TextVQA).
>
> | Threshold | sCLEVR | SciQA | TextVQA | AvgAcc |
> | --------- | ------ | ----- | ------- | ------ |
> | 0         | 57.2   | 92.5  | 72.1    | 73.9   |
> | 0.1       | 56.8   | 92.1  | 71.4    | 73.4   |
> | 0.2       | 55.9   | 91.6  | 70.2    | 72.6   |
>
> - Our setting achieves the highest overall performance. This suggests that there samples where the model has low but non-zero reward provide more effective gradient signals for policy improvement.
> - As $\tau$ increases, performance degrades across tasks. We recommend $\tau = 0$, which maximizes the retention of informative training instances. This ablation has been added to the revised paper's  Appendix D.5.
>
> **Q1:** Robustness to task ordering
> We evaluated RFT's robustness to task ordering:
> **Table 5:** We evaluate two different task orderings on both 3B and 8B models to assess the sensitivity of forgetting mitigation to task sequence.
>
> |         Model          |      Task Order      | Task 1 | Task 2 | Task 3 |  FM  |
> | :--------------------: | :------------------: | :----: | :----: | :----: | :--: |
> | Qwen2.5-VL-3B-Instruct | sCLEVR→SciQA→TextVQA |  57.8  |  92.7  |  72.8  | -0.4 |
> | Qwen2.5-VL-3B-Instruct | TextVQA→SciQA→sCLEVR |  72.8  |  92.1  |  57.8  | -0.3 |
> |  Qwen3-VL-8B-Instruct  | sCLEVR→SciQA→TextVQA |  57.0  |  96.3  |  76.1  | -0.2 |
> |  Qwen3-VL-8B-Instruct  | TextVQA→SciQA→sCLEVR |  76.1  |  96.8  |  55.6  | -0.4 |
>
> As shown in Table 5, RFT consistently maintains low forgetting regardless of task ordering. We have included the task ordering robustness analysis in Appendix D.3 of the revised paper.
>
> **Q3:** Protection of knowledge across different modalities
>
> - Observations from Tables 1 (Text) and 2 (Multimodal) suggest RFT provides universal protection. The forgetting mitigation is effective for both vision-language alignment and pure text reasoning, indicating **no strong modality preference**.
>
>
>
> [1] VLM-R1: A Stable and Generalizable R1-style Large Vision-Language Model
>
> [2] SFT or RL? An Early Investigation into Training R1-Like Reasoning Large Vision-Language Models
>
> [3] Rethinking RL Scaling for Vision Language Models: A Transparent, From-Scratch Framework and Comprehensive Evaluation Scheme
>
> [4] Vision-R1: Incentivizing Reasoning Capability in Multimodal Large Language Models
>
> [5] VL-Rethinker: Incentivizing Self-Reflection of Vision-Language Models with Reinforcement Learning
>
> [6] The Synergy Dilemma of Long-CoT SFT and RL: Investigating Post-Training Techniques for Reasoning VLMs
>
> [7] Qwen2 technical report
>
> [8] Training verifiers to solve math word problems
>
> [9] What disease does this patient have? a large-scale open domain question answering dataset from medical exams
>
> [10] Qwen2. 5-VL technical report
>
> [11] Qwen3 technical report
>
> [12] Revisiting Replay and Gradient Alignment for Continual Pre-Training of Large Language Models
>
> [13] Simple and Scalable Strategies to Continually Pre-train Large Language Models

---

### Official Review · Reviewer_xz8f · 2025-10-28

**Soundness:** 2
**Presentation:** 3
**Contribution:** 3
**Rating:** 6
**Confidence:** 4

**Summary:**

This paper investigates forgetting in two mainstream post-training paradigms: supervised fine-tuning (SFT) and reinforcement fine-tuning (RFT). The authors report that SFT can induce catastrophic forgetting, whereas RFT tends to preserve -- and sometimes even enhance -- general knowledge. They further propose that RFT benefits from an implicit regularization effect in which gradient updates are scaled by reward variance. Building on this, they introduce a rollout-based instance filtering algorithm to improve the stability and efficiency of RFT.

**Strengths:**

1. This paper presents an interesting theoretical claim: RFT mitigates catastrophic forgetting via implicit gradient regularization driven by reward variance.

2. It offers solid experiments supporting that RFT reduces forgetting, and argues the effect is not attributable to KL constraints or chain-of-thought (CoT).

3. The paper is clear writing and overall easy to follow.

**Weaknesses:**

1. My primary concern is, the central theorem lacks convincing empirical support. The proposed rollout-based instance filtering method (RIF-RFT), motivated by the theorem, should improve anti-forgetting via lower reward variance under the theory, yet it underperforms the baseline RFT. The authors note that filtering reduces training data, but this raises a fairness concern: why not match the effective data budget across RIF-RFT and baseline RFT to isolate the effect of variance reduction?

2. The efficiency claim for the filtering method needs stronger evidence. While filtering may yield higher-quality data for subsequent RFT, it introduces a nontrivial overhead from sampling answers from the base LLM -- often a bottleneck for methods like GRPO. We need more solid proof of this method’s efficiency.

I am willing to increase my score if the authors can address the concerns mentioned above.

**Questions:**

1. The related-work discussion can be expanded. For example, [1] also studies SFT vs. RFT and suggests that SFT "directly providing answers to new tasks, without linking them to the model’s existing perceptual abilities through reasoning trajectories, causes the output distribution to shift abruptly". This seems at odds with the paper’s claim that CoT does not mitigate catastrophic forgetting. Could the authors elaborate on the differences between [1] and their work, and clarify the potential contradiction between them?

2. Typo in Eq. (4): the second "$a$" should denote the greedily sampled response and should be a different symbol (e.g., "$\hat{a}$").


[1] Zhang, Z., Dong, Q., Zhang, Q., Zhao, J., Zhou, E., Xi, Z., Jin, S., Fan, X., Zhou, Y., Wu, M., Fu, Y., Ji, T., Gui, T., Huang, X., & Chen, K. (2025, June 30). Why Reinforcement Fine-Tuning Enables MLLMs Preserve Prior Knowledge Better: A Data Perspective

---

> ### Author Response · Authors · 2025-11-24
> **Response to Reviewer xz8f (1/2)**
>
> We sincerely thank the reviewer for the valuable feedback and constructive suggestions. Please kindly find our response to your comments below, and all revisions made to the paper are highlighted in blue for your ease of reference. We hope that our response satisfactorily addresses the questions you raised. Please feel free to let us know if you have any additional concerns or questions.
>
> **W1:** Empirical support and fair comparison
>
> We sincerely thank the reviewer for this insightful suggestion. Following your suggestion for a fair comparison with matched data budgets, we conducted experiments on two tasks (PathVQA → sCLEVR):
>
> **Table 1:** Comparison of RIF-RFT and Data-Matched Random Subsampling
>
> | Method             | PathVQA | sCLEVR |    Avg    |    FM     |
> | ------------------ | :-----: | :----: | :-------: | :-------: |
> | RIF-RFT            |  37.91  | 52.88  | **45.40** | **-0.31** |
> | RFT (Matched Data) |  34.93  | 50.22  |   42.58   |   -0.80   |
>
> * With equal data sizes, RIF-RFT achieves higher overall performance and less forgetting. This suggests that our rollout-based filtering selects more effective training instances compared to random sampling.
>
> Regarding the question why baseline RFT with the whole training set outperforms RIF-RFT, we offer the following explanation:
>
> * This comparison reveals an important insight about the dynamics of policy optimization. RIF-RFT performs one-time pre-filtering before training based on the initial policy's capability. However, the policy evolves throughout the training. Samples that initially yield zero rewards across all rollouts may later produce non-zero rewards as the policy improves. Full-data training retains access to these samples throughout optimization, allowing it to leverage these emerging learning signals. In contrast, RIF-RFT excludes these samples a priori, trading this potential for computational efficiency.
>
> * For samples where all rollouts receive zero reward, the policy gradient term is effectively zero (advantage = 0).  However, the KL penalty term remains active, regularizing the policy toward the reference model.
> * We plan to conduct future exploration of curriculum-based data selection strategies and KL term regularization analysis of different samples. We have revised Section 5.3 to more clearly articulate these observations and the design rationale behind our filtering strategy.

---

> > ### Author Response · Authors · 2025-11-26
> > **Response to Reviewer xz8f (2/2)**
> >
> > **W2:** Efficiency evidence for the filtering method
> >
> > * We provide a concrete efficiency analysis in Table 1. RFT training involves two steps: (1) rollout generation (inference), and (2) policy optimization (backpropagation).
> > * Our filtering step only requires inference (Step 1), which is much cheaper than the full training loop (Step 1 + Step 2).
> > * As shown below, filtering introduces a negligible overhead (<2%) but reduces the training time by ~44%., while maintaining competitive performance. This confirms RIF-RFT improves both computational efficiency and sample efficiency.
> >
> > **Table 2:** We report wall-clock time (hours) on 8×H800 GPUs, showing both filtering overhead and training time for each task.
> >
> > | Dataset   | GRPO  | RIF-RFT Training | RIF-RFT Filtering | RIF-RFT |
> > | --------- | ----- | ---------------- | ----------------- | ------- |
> > | ScienceQA | 6.4   | 5.2              | 0.13              | 5.33    |
> > | TextVQA   | 30.9  | 13.8             | 0.31              | 14.11   |
> > | VizWiz    | 19.5  | 8.0              | 0.20              | 8.20    |
> > | GQA       | 72.6  | 48.6             | 0.50              | 49.10   |
> > | Geo.      | 2.3   | 0.6              | 0.02              | 0.62    |
> > | PathVQA   | 15.4  | 5.6              | 0.15              | 5.75    |
> > | sCLEVR    | 6.7   | 3.4              | 0.11              | 3.51    |
> > | **Total** | 153.8 | 85.2             | 1.42              | 86.62   |
> >
> > We have included this efficiency analysis in Appendix D.4 of the revised paper.
> >
> > **Q1:** Clarification on related work [1] and and the role of CoT
> >
> > We thank the reviewer for bringing this insightful related work [1] to our attention. After carefully reading, we find that [1] provides valuable complementary insights. Here we clarify the relationship between our work and theirs:
> >
> > * Different setups lead to different insights. [1] investigates how different data types within SFT affect forgetting (Non-Reasoning vs. Reasoning data from different sources). Their key finding is adding reasoning trajectories to SFT reduces catastrophic forgetting. We investigate what drives RFT's superior forgetting mitigation. Our key finding is RFT inherently preserves knowledge with or without explicit reasoning. These different research angles provide complementary perspectives on the forgetting problem.
> > * The apparent contradiction about CoT's importance is clarified when we distinguish between **data source** and **data format**. [1]'s findings highlights both the data source (model's own distribution) and format (reasoning) contribute to reducing forgetting in SFT. Our experiments demonstrate that RFT's primary forgetting resistance operates independently of reasoning format, though CoT provides additional benefits. Both studies show that reasoning trajectories can help reduce forgetting, but through different mechanisms.
> > * Our statement that *CoT is not the primary factor of forgetting mitigation* specifically refers to the **RFT paradigm**, where the core mechanisms operate independently of reasoning format. We fully agree with [1] that reasoning trajectories are beneficial for improving SFT's stability.
> > * Both works together provide practical guidance: [1] shows how to make SFT more robust through better data design, while we demonstrate that RFT offers stronger protection against forgetting.
> >
> > We have revised our paper (in Related Works) to cite [1], acknowledge its contributions, and clarify how our findings complement theirs regarding data format and learning paradigm.
> >
> > **Q2:** Typo correction in Equation 4
> >
> > Thank you for catching this error. We have corrected Equation 4 in the revised paper.
> >
> > [1] Why Reinforcement Fine-Tuning Enables MLLMs Preserve Prior Knowledge Better: A Data Perspective

---

### Official Review · Reviewer_gd9L · 2025-10-30

**Soundness:** 3
**Presentation:** 4
**Contribution:** 3
**Rating:** 8
**Confidence:** 5

**Summary:**

This paper presents a compelling study on continual post-training (CPT) of large foundation models, comparing two fine-tuning paradigms: supervised fine-tuning (SFT) versus reinforcement fine-tuning (RFT). The authors conduct extensive experiments on a sequence of seven diverse multimodal tasks using a 7B-parameter vision-language model (Qwen2.5-VL-7B-Instruct) as the base. The key finding is that RFT dramatically outperforms SFT in mitigating catastrophic forgetting: when tasks are learned one after another, SFT suffers severe forgetting of earlier tasks, whereas RFT inherently preserves prior task knowledge, achieving performance on old tasks comparable to an ideal multi-task training baseline. Remarkably, RFT even maintains or improves the model’s general knowledge (evaluated on broad benchmarks like MMMU and MMLU-Pro), whereas SFT significantly degrades general capabilities. The study identifies that this superior knowledge retention of RFT is not primarily due to common tricks like KL-divergence regularization or chain-of-thought prompting, but rather stems from an implicit regularization effect in the RFT paradigm itself. To address the practical challenge that vanilla RFT can be less sample-efficient and stable, the authors propose a simple yet effective addition: a Rollout-based Instance Filtering (RIF) algorithm. RIF prunes “incompetent” training instances (those where the current policy fails to produce any rewarding output) before applying RFT, which significantly improves training stability and efficiency without sacrificing performance. Overall, the paper’s contributions include: (1) the first comprehensive analysis demonstrating RFT’s natural advantage in preserving both task-specific and general knowledge during continual fine-tuning, (2) a theoretical insight that RFT’s use of reward signals induces an implicit, data-dependent regularization that mitigates forgetting better than explicit techniques, and (3) the introduction of the RIF-RFT algorithm to enhance RFT’s efficiency while retaining its robustness to forgetting

**Strengths:**

Originality & Importance: The paper addresses a previously overlooked question – whether reinforcement learning-based fine-tuning could fundamentally improve continual learning – making it a fresh and notable contribution. This exploration of the fine-tuning paradigm itself (as opposed to add-on techniques) is highly original. It targets the important problem of catastrophic forgetting in lifelong learning for foundation models, which has broad relevance for deployed AI systems. Empirical Results – Effectiveness of RFT: The experimental results are impressive and persuasive. RFT nearly eliminates catastrophic forgetting on a diverse set of seven tasks, matching the performance of multi-task training without using any memory or regularization tricks. It also preserves general knowledge and even slightly improves overall capabilities of the base model – a striking outcome since typical fine-tuning often degrades general performance. The magnitude of improvement of RFT over SFT (e.g. retaining old task accuracy ~95% vs dropping to ~70% for SFT, in some cases) demonstrates clear superiority. These strong empirical findings support the paper’s claims and indicate high practical utility. Theoretical Rigor and Insight: The authors provide a sound theoretical explanation for RFT’s advantages. They introduce a forgetting risk metric and prove a bound (Theorem 5.2) showing RFT’s expected forgetting risk is scaled down by reward variance compared to SFT. This insight reveals an implicit regularization effect: RFT’s stochastic policy updates naturally avoid large detrimental shifts in parameters important to past tasks. The theory is novel and aligns with the empirical observations, adding significant credibility and understanding to the results. Comprehensive Analysis and Ablations: The paper is thorough in its analysis. It evaluates multiple RFT algorithms (e.g. GRPO, ReMax, etc.) and ablates key factors: for instance, showing that removing the KL penalty or chain-of-thought does not erase RFT’s forgetting mitigation. This rules out alternative explanations and solidifies that the benefit comes from the core RFT mechanism. The authors also measure both task-specific retention and general knowledge retention, providing a holistic view of the model’s performance. The inclusion of RIF-RFT with results (Table 5) demonstrates the authors’ depth of understanding: they not only identified RFT’s strength but also tackled its weakness (instability on some samples) with a validated solution. Overall, the experimental section leaves little doubt about the conclusions due to its breadth and careful design. Clarity and Presentation: The paper is well-written and organized, making it easy to follow the complex subject matter. The motivation is clearly stated, related work is well-situated, and the contributions are clearly itemized. Figures and tables are effectively used to illustrate key points (e.g. a graph showing SFT vs RFT forgetting dynamics), which aids understanding. The clarity of presentation means the ideas and results are accessible to readers, increasing the impact of the work. Relevance and Impact: The work has strong relevance to multiple areas: continual learning, large-scale model fine-tuning, and reinforcement learning. Its findings are likely to influence future research and practice, as they suggest a paradigm shift (incorporating RL objectives for better retention). The method is also practical – e.g., RIF makes RFT feasible by reducing data needs – which could encourage real-world adoption. The paper thus excels not only academically (with insight and rigor) but also in potential real-world impact, aligning well with ICLR’s interests in advancing learnability of AI systems in dynamic settings.

**Weaknesses:**

Computational Complexity: RFT requires rollouts and policy optimization, making it more computationally intensive and sensitive to hyperparameters than standard SFT. While RIF improves efficiency, real-world deployments may still face engineering challenges. Evaluation Scope: The study focuses on one 7B multimodal model and QA-style tasks. Broader validation across domains, tasks, and model scales would strengthen the generality claims. Reward Dependence: RFT assumes access to clear reward signals. In tasks lacking binary correctness (e.g., subjective or open-ended outputs), reward design could be non-trivial, reducing RFT’s immediate applicability. Initial Model Competence: RFT performs best when the base model has some proficiency on new tasks. In cold-start scenarios with zero initial success, the reward sparsity could limit RFT’s learning. RIF helps mitigate this but may discard too much data in such cases. Limited Baseline Comparisons: While the contrast to vanilla SFT is strong, the paper could benefit from including other continual learning baselines (e.g., SFT with replay or regularization) to further isolate the benefit of the RFT paradigm. These limitations are either practical considerations or natural directions for future research and do not detract from the core strength of the paper.

**Questions:**

Generality to Other Settings: How well do you expect the advantages of RFT to transfer to other model types and domains? For example, have you considered or attempted applying RFT-based continual learning on a pure text large language model, or a smaller model with less prior knowledge? It would be insightful to know if RFT’s implicit regularization still shines in those cases, or if there are any domains where SFT might remain competitive. Reward Function Design: Could you clarify how the reward signals were defined for your tasks, and whether any reward shaping or learned reward models were needed? Since the tasks seem to have ground-truth answers (e.g. classification or QA), was the reward simply 1 for a correct answer and 0 for incorrect, or something more nuanced (partial credit, etc.)? Understanding this would help gauge how easily RFT can be applied to new tasks. Additionally, if a task had a more subjective or continuous evaluation metric, do you foresee any challenges in applying RFT there? RFT in Low-Performance Regimes: How does RFT perform when the model’s initial success on a new task is very low, and how effective is RIF in that scenario? The paper introduces RIF to handle “incompetent” samples – for a new task where the model almost never produces a correct output initially, RIF would filter out many samples. Does this risk filtering out too much data and hindering learning new knowledge? In other words, is there a point at which a task is so difficult that a bit of supervised signal might be needed to kickstart learning before RFT takes over? Any discussion on this or empirical observation (even anecdotal) would be appreciated to understand the limits of RFT’s applicability. Combining RFT with Other CL Techniques: Have you considered if combining RFT with traditional continual learning techniques (like a small experience replay buffer or weight regularization) would further improve performance, or is RFT alone already near-optimal? Your results suggest RFT alone is as good as multi-task training, which is remarkable. But for completeness, it would be interesting to know if adding even minor rehearsal of old tasks (or other CL tricks) on top of RFT yields any marginal gains or if it’s unnecessary. This could help practitioners decide how to allocate resources (e.g., focus purely on RFT or also maintain some memory of past data). (Feel free to address these during the rebuttal if time permits – they are mostly aimed at understanding the breadth of applicability and practical considerations of the proposed approach.)

---

> ### Author Response · Authors · 2025-11-24
> **Response To Reviewer gd9L (1/2)**
>
> We sincerely appreciate your thorough and insightful review. Your comprehensive analysis has helped us identify important areas for clarification and improvement. We are grateful for your recognition of our work's originality in exploring the fundamental learning paradigm for continual post-training, and your valuable suggestions for future directions.
>
> Below we address each concern. Some experimental results have been updated in the revised paper, and any modifications made to the paper are highlighted in blue for your convenience. Please kindly let us know whether you have any further concerns.
>
>
>  **W1:** Computational Complexity
>
> - We acknowledge that RFT generally requires more computational resources due to the rollout generation phase. However, we have conducted a comprehensive efficiency analysis comparing standard GRPO and our proposed RIF-RFT.
> - As shown in Table 1 below, RIF-RFT significantly reduces the total wall-clock time with negligible filtering overhead. This makes the method highly practical for deployment.
> - Regarding hyperparameter sensitivity, we find RFT to be surprisingly robust; we use a unified set of hyperparameters across all tasks, whereas SFT requires task-specific learning rate tuning to balance plasticity and stability.
>
> **Table 1:** Wall-clock time (hours) comparison.
>
> | Dataset    | GRPO  | RIF-RFT Training | RIF-RFT Filtering |  RIF-RFT  |
> | ---------- | ----- | :--------------: | :---------------: | :-------: |
> | ScienceQA  | 6.4   |       5.2        |       0.13        |   5.33    |
> | TextVQA    | 30.9  |       13.8       |       0.31        |   14.11   |
> | VizWiz     | 19.5  |       8.0        |       0.20        |   8.20    |
> | GQA        | 72.6  |       48.6       |       0.50        |   49.10   |
> | Geometry3k | 2.3   |       0.6        |       0.02        |   0.62    |
> | PathVQA    | 15.4  |       5.6        |       0.15        |   5.75    |
> | sCLEVR     | 6.7   |       3.4        |       0.11        |   3.51    |
> | Total      | 153.8 |       85.2       |       1.42        | **86.62** |
>
> **W2 & Q1:** Evaluation Scope and Generality
>
> - We appreciate this suggestion. We have extended our experiments to validate generality across different architectures (text-only models) and model scales (3B and 8B).
> - First, we evaluate Qwen2.5-7B-Instruct on GSM8K (Math) and USMLE (Medical). As shown in Table 2, RFT consistently mitigates forgetting, proving the benefit is not limited to MLLMs.
> - We also evaluate Qwen2.5-VL-3B and Qwen3-VL-8B. As shown in Table 3, RFT maintains near-zero forgetting across scales.
>
> **Table 2**: Comparison on Qwen2.5-7B-Instruct.
>
> | Method | Task Order  | GSM8K | USMLE | FM |
> | :----: | :---------: | :---: | :---: | :----------------: |
> |  RFT   | GSM8K→USMLE | 84.2  | 62.3  |        -1.8        |
> |        | USMLE→GSM8K | 85.1  | 60.7  |        -1.2        |
> |  SFT   | GSM8K→USMLE | 71.3  | 58.2  |       -10.4        |
> |        | USMLE→GSM8K | 82.4  | 49.6  |        -8.7        |
>
> **Table 3**: Performance across different model scales (Task Order: sCLEVR→ScienceQA→TextVQA).
>
> |       Model       | Method | sCLEVR | ScienceQA | TextVQA | FM |
> | :---------------: | :----: | :----: | :-------: | :-----: | :----------------: |
> | Qwen2.5-VL-3B |  RFT   |  57.8  |   92.7    |  72.8   |      **-0.4**      |
> |                   |  SFT   |  51.5  |   92.3    |  67.6   |        -4.4        |
> |  Qwen3-VL-8B  |  RFT   |  57.0  |   96.3    |  76.1   |      **-0.2**      |
> |                   |  SFT   |  48.2  |   91.5    |  68.9   |        -7.1        |

---

> > ### Author Response · Authors · 2025-11-24
> > **Response To Reviewer gd9L (2/2)**
> >
> > **W3 & Q2**: Reward Design and Dependence
> >
> > - We use binary correctness with format checking (Eq. 6). We do not use partial credit or continuous rewards in the current experiments
> > - Regarding subjective tasks, the main additional challenge in such settings is **reward design and noise** (how to obtain reliable scalar evaluations), which is common to any RFT method and orthogonal to our continual post‑training analysis.
> >
> > **W4 & Q3:** Performance in Low-Competence Regimes
> >
> > * While our experiments start from models with non-zero capabilities, RFT does not require high initial competence. It only requires the model to have a non-zero probability of generating a correct response during exploration. As supported by recent work [1], RFT can succeed from low initial performance by amplifying these sparse success signals. In contrast, if a model has strictly zero capability, a brief SFT warm-up would help, which is a standard practice in RLHF.
> >
> > **W5 & Q4:** Comparison with Other CL Techniques
> >
> > - We appreciate this suggestion. And we have conducted a comparison with Experience Replay (ER), a strong CL baseline. We implemented SFT with a 25% replay ratio.
> > - Table 4 shows that while ER reduces forgetting compared to SFT, it still lags behind RFT and introduces storage overhead. This confirms RFT's inherent superiority in knowledge retention.
> >
> > **Table 4:** RFT and SFT+ER Comparison  on Qwen2.5-VL-3B.
> >
> > | Method  | sCLEVR | ScienceQA | TextVQA | FM |
> > | ------- | ------ | --------- | ------- | ------------------ |
> > | SFT     | 51.5   | 92.3      | 67.6    | -4.4               |
> > | SFT+ ER | 53.2   | 92.1      | 64.5    | -2.8               |
> > | RFT     | 57.8   | 92.7      | 72.8    | **-0.4**           |
> >
> > [1] ReFT: Reasoning with Reinforced Fine-Tuning

---

### Author Response · Authors · 2025-12-02
**Summary of Rebuttal**

We sincerely thank all reviewers for their constructive feedback. Below we summarize the main points from the discussion.

**Acknowledged strengths:**

We are encouraged that reviewers recognized several merits of our work, including the novelty of exploring the fundamental learning paradigm (SFT vs. RFT) for continual post-training (gd9L, xz8f, wgFr), the impressive empirical results demonstrating RFT's forgetting mitigation (gd9L), the theoretical insights on implicit regularization (gd9L, xz8f, wgFr), and the comprehensive analysis with thorough ablations and clear presentation (gd9L, xz8f).

**Additional experiments:**

- We conducted experiments across different model scales (3B, 7B, 8B) and domains (text and multimodal), showing that RFT maintains forgetting mitigation consistently. (wgFr W1 & Q2, gd9L W2)

- We compared RFT with Experience Replay, showing that RFT achieves better forgetting mitigation without extra memory buffers. (wgFr W2, gd9L W5)
- Fair comparison between RIF-RFT and data-matched random subsampling, validating our filtering strategy. (xz8f W1)
- Experiments with different task orderings. (wgFr Q1)
- Ablation studies on filtering threshold. (wgFr W3)

**Clarifications:**

- We included detailed training time analysis of RIF-RFT. (xz8f W2, gd9L W1)

- Discussion on how our work complements concurrent work. (xz8f Q1)

- We corrected the typo in Equation 4. (xz8f Q2)

All revisions are highlighted in blue in the updated manuscript. We believe our responses have addressed all reviewers' concerns.

---

### Note · Program_Chairs · 2026-01-17
**Submission Desk Rejected by Program Chairs**

The following references in this submission do not refer to real documents and/or have major errors in bibliographic information:

 Jing Shao, Kun Li, Wentao Dong, et al. Deepseekmath: A multimodal multitask benchmark for mathematical reasoning. arXiv preprint arXiv:2406.01297, 2024.